# Microbial Activity in Subterranean Ecosystems: Recent Advances

**Tamara Martin-Pozas [1,†]** , **Jose Luis Gonzalez-Pimentel [2,†]** , **Valme Jurado [3]** , **Soledad Cuezva [4]** ,
**Irene Dominguez-Moñino [3]** , **Angel Fernandez-Cortes [5]** , **Juan Carlos Cañaveras [6]** ,
**Sergio Sanchez-Moral [1]** and **Cesareo Saiz-Jimenez [3,*]**

[1] Museo Nacional de Ciencias Naturales (MNCN-CSIC), 28006 Madrid, Spain;
tmpozas@mncn.csic.es (T.M.-P.); ssmilk@mncn.csic.es (S.S.-M.)
[2] Laboratorio HERCULES, Universidade de Evora, 7000-809 Evora, Portugal; pimentel@irnas.csic.es
[3] Instituto de Recursos Naturales y Agrobiologia (IRNAS-CSIC), 41012 Sevilla, Spain;
vjurado@irnase.csic.es (V.J.); idominguez@irnase.csic.es (I.D.-M.)
[4] Departamento de Geología, Geografía y Ciencias Ambientales, Universidad de Alcala de Henares,
28805 Madrid, Spain; soledadcuezva@uah.es
[5] Departmento de Biología y Geología, Universidad de Almeria, 04120 Almeria, Spain; acortes@ual.es
[6] Departamento de Ciencias de la Tierra, Universidad de Alicante, 03080 Alicante, Spain; jc.canaveras@ua.es
* Correspondence: saiz@irnase.csic.es
† These authors contributed equally to this study.

**Abstract:** Of the several critical challenges present in environmental microbiology today, one is the assessment of the contribution of microorganisms in the carbon cycle in the Earth-climate system. Karstic subterranean ecosystems have been overlooked until recently. Covering up to 25% of the land surface and acting as a rapid $CH_4$ sink and alternately as a $CO_2$ source or sink, karstic subterranean ecosystems play a decisive role in the carbon cycle in terms of their contribution to the global balance of greenhouse gases. Recent data indicate that microbiota must play a significant ecological role in the biogeochemical processes that control the composition of the subterranean atmosphere, as well as in the availability of nutrients for the ecosystem. Nevertheless, there are still essential gaps in our knowledge concerning the budgets of greenhouse gases at the ecosystem scale and the possible feedback mechanisms between environmental-microclimatic conditions and the rates and type of activity of microbial communities in subterranean ecosystems. Another challenge is searching for bioactive compounds (antibiotics) used for treating human diseases. At present, there is a global health emergency and a strong need for novel biomolecules. In recent decades, great research efforts have been made to extract antibiotics from marine organisms. More recently, caves have been receiving considerable attention in search of novel antibiotics. Cave methanotrophic and heterotrophic bacteria are producers of bioactive compounds and may be potential sources of metabolites with antibacterial, antifungal or anticancer activities of interest in pharmacological and medical research, as well as enzymes with a further biotechnological use. Here we also show that bacteria isolated from mines, a still unexplored niche for scientists in search of novel compounds, can be a source of novel secondary metabolites.

**Keywords:** karst; methane; carbon dioxide; greenhouse gases; methanotrophy; cave bacteria; bioactive compounds

## 1. Introduction

Karst is the term used to describe terrains underlain by soluble rock and characterized by the occurrence of caves, sinkholes, sinking streams, and an assortment of other landforms carved on the

bedrock. Shallow karst ecosystems cover up to 25% of the Earth's land surface [1] and differ from the surface environments because of their limited energy and available nutrients.

Caves, in general, are characterized by a constant temperature, humidity, and high carbon dioxide ($CO_2$) concentration the year round, as well as absence of light and scarcity of nutrients. Microorganisms occupy all the niches of the biosphere, including the subsurface, as a part of the critical zone, the heterogeneous near surface environment in which complex interactions involve rock, soil, water, air, and living organisms [2].

Earth's subsurface contains an active microbiota colonizing rock surfaces. In this environment, microorganisms are forced to adapt their metabolism for surviving in extreme conditions, and the low input of carbon, nitrogen and phosphorus as well as the chemical composition of the rock has a direct impact on the community diversity. In fact, one of the main reservoirs of microbial life, even at great depths, where life is not dependent on solar energy and photosynthesis for its primary energy supply is the terrestrial subsurface [3].

The colonization of substrates in caves is not homogeneous. Microorganisms colonize speleothems, host rock, detrital sediments, and/or speleosols with different compositions (clays, carbonate minerals, etc.) and/or textures (crystal habit, grain size, permeability, etc.). Microbial colonization is ultimately a complex and dynamic process that is determined and controlled by physicochemical properties (temperature, pH, redox potential, salinity) and biochemical factors (bioreceptivity, nutrient or electron acceptor availability, carbon, nitrogen and phosphorous concentrations, etc.) [4]. Therefore, the collective metabolic processes of microorganisms are decisive in the biogeochemical cycles of the biosphere: C and N fixation, $CH_4$ metabolism, S oxide reduction, etc.

It is well-known that dissolution and precipitation of carbonates are the main processes involved in the mobilization of carbon in subterranean environments. Cave microorganisms are able to induce the precipitation of carbonates, via biomineralization processes [5] and also dissolution processes due to the excretion of acids [6]. There is a wide array of literature on the study of bio-induced mineral formations in subterranean environments [7] and on the microbial–rock interaction related to the $CO_2$ uptake or release processes [8]. In this context, previous studies have confirmed that *Actinobacteria* biofilms developing on cave walls promote uptake of $CO_2$, dissolve the rock, and produce calcite crystals in periods of lower humidity and/or $CO_2$ [8]. However, the interactions of microbes with the air–water–rock interfaces in subterranean ecosystems and the biological mechanisms by which microorganisms adjust to new environments or changes in their current environment are poorly understood.

Low energy subsurface environments are uniquely positioned for examining minimum energetic requirements and adaptations for chemolithotrophic life and become a suitable environment to study the origins of life on Earth and may also serve as analogs to explore subsurface life in extraterrestrial bodies [9]. Furthermore, the microbiota from shallow subsurface environments (karst cavities, lava tubes) are becoming a target of increasing interest in different research fields, including biodiversity [10], mineral formation and dissolution [7], cultural and natural heritage conservation [11], and paleoclimatology [12]. In addition, other important uses of microorganisms are the production of bioactive compounds valuable for medicine and enzymes for bioremediation [13].

The extensive literature about microbial diversity and activity of cave microorganisms has been reviewed by many authors. The books "Microbial Life of Cave Systems" by Engel [14] and "Cave Ecology" by Moldovan et al. [15] are a rich source of information. In addition, other book chapters and review articles are relevant [16–21]. Because of the comprehensive scope of the literature on this topic, for this review we have selected two emerging research topics representing recent advances in environmental microbiology: (1) the control of greenhouse gas fluxes by cave microorganisms, and (2) the search of antibiotics produced by subsurface bacteria.

## 2. The Control of Greenhouse Gas Fluxes by Cave Microorganisms

Global changes in the Earth's climate and its relationship to the increasing concentration of greenhouse gases (GHGs) in the atmosphere has received special attention since the last quarter of the 20th century. Etiope and Klusman [22] reported that the major sources for atmospheric methane ($CH_4$) budget derive from the natural processes in the biosphere (modern microbial activity) and from fossil, radiocarbon-free $CH_4$ emission, estimated at approximately 20% of atmospheric $CH_4$, which is due to and mediated by anthropogenic activity. However, this estimation is higher than the estimates from statistical data of $CH_4$ emission from fossil fuel and related anthropogenic sources. For these authors, geologic sources are more than enough to provide the amount of $CH_4$ required to account for the suspected missing source of fossil $CH_4$. In addition, Etiope and Lollar [23] distinguished between biotic and abiotic $CH_4$, the latter produced in magmatic processes (volcanoes and high-temperature active hydrothermal vents) and postmagmatic processes at lower temperatures (gas–water–rock interactions).

A better understanding of the carbon cycle in the Earth-climate system is nowadays a crucial knowledge gap. The main research efforts are focused on identifying and characterizing all possible sources, reservoirs, and sinks of GHGs, mainly $CO_2$ and $CH_4$, in order to more accurately calculate the budgets, especially in the carbon cycle [24]. This issue is critical to understand the effects of changes in the carbon cycle on Earth's climate, and to assess the level of effort required in order to adapt and mitigate climate change.

The interactions between geological, microbiological, and chemical processes are responsible for the physical-chemical properties of the atmosphere and especially for changes in its composition. Caves and other shallow vadose environments are populated by methanotrophic microorganisms and thus represent a $CH_4$ sink. This subterranean $CH_4$ sink is largely overlooked in the scientific literature. Understanding how cave microbiomes influence the systems in which they inhabit is proving to be an exceptional research challenge [25].

Methane is consumed from the atmosphere by methanotrophs in forests, grasslands, paddy, and other unsaturated soils, which represent the major terrestrial sinks. Environmental $CH_4$ oxidation by bacteria is mainly carried out by *Gammaproteobacteria*, *Alphaproteobacteria,* and *Verrucomicrobia* [26], though there is also recent evidence for methanotrophy in *Rokubacteria* [27].

The presence of methanotrophic bacteria in caves has been widely studied in Movile Cave, Romania, by using isolation techniques, $^{13}CH_4$-labelling, and $^{13}C$-DNA analysis, and the significant importance to the ecosystem development and primary productivity has been remarked upon [28–31]. Evidence of the occurrence of methanotrophs has also been found in other caves [10,32,33]. However, in these studies the microorganisms were not related with the sink of GHGs in caves.

Specific studies, both on the environmental-driven controls on microbial activity and, in turn, on the microbial role in composition changes of natural subterranean ecosystems, constitute a new research area of the highest potential with a pool of questions to solve. The starting hypothesis was that the subterranean microbiome plays a significant ecological role in the biogeochemical processes controlling the composition of the underground atmosphere, as well as in the availability of nutrients for the rest of the ecosystem's biota.

Fernandez-Cortes et al. [34] evidenced for the first time that cave ecosystems act as effective natural sinks of atmospheric $CH_4$ on seasonal and daily scales and this phenomenon may thus be relevant on a global scale in terms of its contribution to the global balance of GHGs. The potential methanotrophy in four Spanish caves was assessed by tracking the presence of methane-oxidizing bacteria using the particulate methane monooxygenase gene *pmoA*, which is a phylogenetic marker for identifying methanotroph-specific DNA sequences in the environment [35]. The study revealed the presence of the proteobacteria *Methylocapsa aurea*, *Methylomicrobium album*, *Methylococcus capsulatus,* and methanotrophs of the K1-1 and K3-16 groups in samples from Altamira, Sidron, and Ojo Guareña caves, mainly in locations where $CH_4$ usually reaches concentrations near to the atmospheric background levels. These soil bacteria oxidize the atmospheric $CH_4$ [36].

However, the analyses did not detect methanotrophs in remote subterranean locations or poorly ventilated caves, such as Castañar de Ibor Cave, where $CH_4$ is absent or present in minimal concentrations (below the accuracy threshold) throughout the year. Fernandez-Cortes et al. [34] suggested that complete consumption of $CH_4$ was favored in the subsurface atmosphere under near vapor-saturation conditions without significant intervention of methanotrophic bacteria. This led to the assumption that $CH_4$ oxidation was induced by ions and ●OH generated by the radioactive decay of radon ($^{222}$Rn). In fact, one of the important ●OH sources in cave air may be from radioactive $^{222}$Rn decay [34]. However, further research verified that the mechanism of $CH_4$ consumption was seasonally changing and methane-oxidizing bacteria were primarily responsible for the widespread observations of $CH_4$ depletion in subterranean environments, discarding any evidence of radiolysis contribution [37–39].

Schimmelmann et al. [37] tested, in controlled laboratory experiments, whether radiolysis could rapidly oxidize $CH_4$ in sealed air with different relative humidity and elevated levels of radiation from Rn isotopes. No evidence of $CH_4$ oxidation by radiolysis was found. On the contrary, a rapid loss of $CH_4$ was found when moist soil in the absence of Rn was added to the container. This was consistent with the presence of methane-oxidizing bacteria, which were responsible for the widespread observations of $CH_4$ depletion in subterranean environments.

Since the pioneering work of Fernandez-Cortes et al. [34], a few authors, based on studies in caves from Australia, the USA, Vietnam, and Spain, additionally supported $CH_4$ oxidation by methanotrophic bacteria [38–42].

Webster et al. [38] reported that the concentrations and stable isotopic compositions of $CH_4$, $CO_2$, and Rn in cave air overlapped and diverged from those of the atmosphere, as the majority of cave air samples were depleted in $CH_4$ and enriched in $CO_2$ relative to the local atmosphere. These differences indicate that atmospheric and internal cave processes influenced the composition of cave air. Therefore, the authors, on the basis of $CH_4$ concentrations, $\delta^{13}C_{CH4}$, and $\delta^2H_{CH4}$ values measured in 33 caves in the USA and three caves in New Zealand, suggested that microbial methanotrophy within caves is the primary $CH_4$ consumption mechanism. Furthermore, the stable isotopic composition of $CH_4$ in the studied caves suggested that, in addition to atmospheric $CH_4$, at least two additional $CH_4$ sources were present in some caves: $CH_4$ produced from acetate fermentation, and from $CO_2$ reduction, processes occurring over a wide scale in the environment.

Lennon et al. [39] also proposed that biological processes, largely oxidation by methanotrophic bacteria, cause a depletion of $CH_4$ in caves. They conducted a field mesocosm experiment to test whether or not microbial methanotrophy has the potential to act as a daily sink for $CH_4$ in two fairly well-ventilated Vietnamese caves with low Rn concentrations (75–115 Bq/m$^3$), temperatures of 19–21 °C, and relative humidity ranging between 85 and 95%, depending on the airflow and location within the cave. The data suggested that biological processes have the potential to deplete atmospheric levels of $CH_4$ (~2 ppmv) via methanotrophy on a daily basis, as an 87% reduction in $CH_4$ concentrations was observed.

It appears that $CH_4$ depletion is a seasonal phenomenon, as reported by several authors. Fernandez-Cortes et al. [34] found significant seasonal and even daily variations in the gas composition of cave air, which involves the exchange of large amounts of other GHGs, in addition to $CO_2$(g), with the lower troposphere. Waring et al. [40] performed a continuous 3-year record of $CH_4$ and other trace gases in an Australian cave and found a seasonal cycle of extreme $CH_4$ depletion, from ambient ~1775 ppb to near zero during summer and to ~800 ppb in winter.

Ojeda et al. [41] found methanotrophic bacteria from the families *Methylococcaceae* (*Gammaproteobacteria*) and *Methylocystaceae* (*Alphaproteobacteria*) in 67% of the samples collected in Nerja Cave, Spain. In a recent innovative research, Cuezva et al. [42] confirmed that microbial action in caves plays a crucial role in the processes of production, consumption and storage of GHGs ($CO_2$ and $CH_4$) and largely determines the strong variations of these major GHGs in natural underground ecosystems. This study was developed in three Spanish caves (Pindal, Castañar de Ibor and La Garma) as a first

approach to systematically characterize the role of cave sediments in the production and transport of $CO_2$ and $CH_4$ in the subterranean environment.

Monitoring and sampling for more than two years in La Garma Cave showed that during the stages with greater ventilation, air circulates daily and there is a continual contribution of external air to the cave, which has lower $CO_2$ content and $CH_4$ levels close to the atmospheric background. Therefore, $CH_4$ depletion rises with slight changes in $CO_2$. Conversely, in stages with a low ventilation rate, $CO_2$ reaches high concentrations in the cave because air exchange with the external atmosphere is negligible. Thus, the removed $CH_4$ is not rapidly replenished. As a result, $CH_4$ depletion rate tends to become negligible as the $CO_2$ content of cave air rises (Figure 1a).

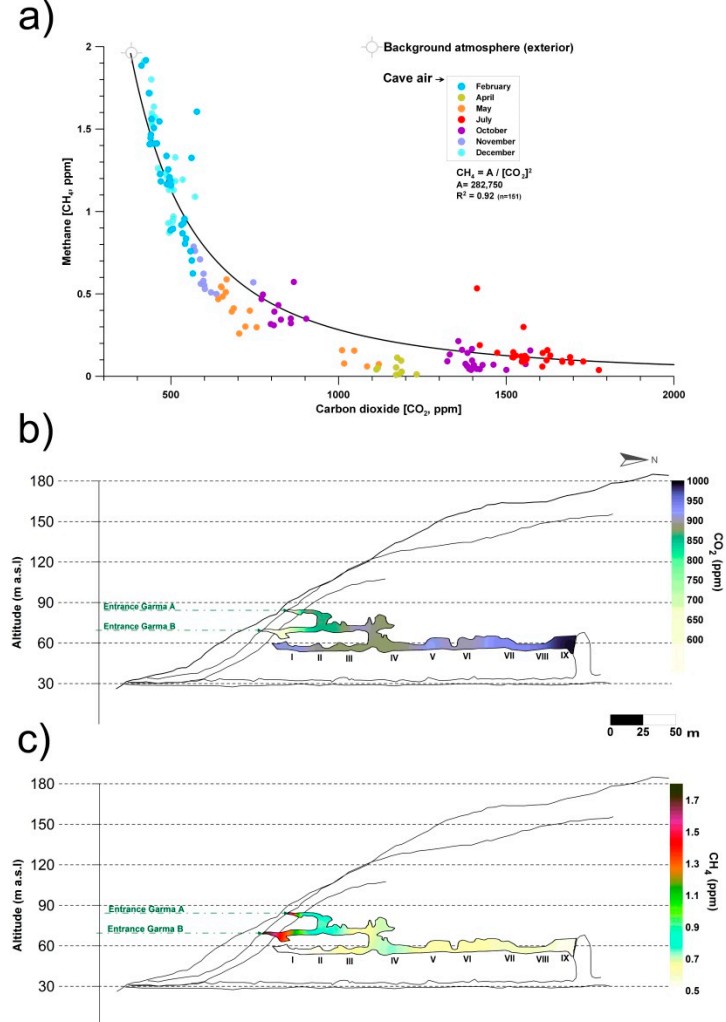

**Figure 1.** (**a**) Monthly co-variations in the concentrations of carbon dioxide ($CO_2$) and methane ($CH_4$) in La Garma Cave (Cantabria, northern Spain), a dynamically ventilated cave. (**b**) Spatial distribution of average concentrations of $CO_2$. (**c**) Spatial distribution of average concentrations of $CH_4$. Data from October 2014 to July 2017.

Figure 1b,c shows the spatial distribution of air $CO_2$ and $CH_4$ concentrations, respectively, in the air from La Garma Cave. Data for each contour map correspond to mean values from a set of bimonthly spot air samplings, conducted from October 2014 to July 2017, in a pre-established network of 11 points covering up to three levels of cave passages along an altitude gradient.

The average $CO_2$ and $CH_4$ concentrations of cave air were 894 and 0.65 ppm, respectively. Both GHGs depend on the rate of cave air exchange with the local atmosphere, which is controlled by climate-driven processes (primarily advection), and it is a very good indicator of the levels of

matter and energy exchange with the exterior, showing the isolated areas and those with a prevailing connection with the exterior. Thus, a remarkable spatial pattern is distinguished; the highest average values of $CO_2$ concentration and the lowest $CH_4$ were found in the sectors of the lower gallery furthest from the main cave entrances (Garma A and Garma B, Figure 1b,c). Therefore, these cave maps with the contoured $CO_2$ and $CH_4$ levels reveal the importance of cave morphology in complex subterranean systems which control the gaseous composition of cave air, particularly in terms of gas variations due to the occurrence of elevation changes, multiple entrances or presence of dead-end passages. In the case of $CH_4$, its average concentration decreased drastically below 0.7 ppm from the connection of the intermediate gallery with the lower gallery and was practically null (<0.5 ppm) in the most distant sectors of the cave entrances (Figure 1c). This $CH_4$ pattern results from a decreasing percentage of mixing with the exterior and, consequently, a more effective methanotrophic activity of bacterial origin.

Cuezva et al. [42] are developing seasonal campaigns for $CH_4$ and $CO_2$ daily fluxes with continuous monitoring by a closed chamber-based gas exchange system (LI-COR Automated Soil Gas Flux System), in conjunction with a compatible Gasmet Fourier Transform Infrared (FTIR) gas analyzer and combined with $\delta^{13}C$ geochemical tracing by cavity ring-down spectroscopy (CRDS) to understand the underlying mechanisms in cave sediments. Moreover, an autonomous piece of equipment monitored the main microenvironmental parameters of the local subsurface-soil-atmosphere system. Preliminary results showed net $CO_2$ emissions from cave sediments resulting from respiration by chemolithotrophic microorganisms. The results also revealed simultaneous net $CH_4$ uptake from cave sediments on a daily scale, with no significant level of variations along the day (Figure 2). Anaerobic oxidation of $CH_4$ coupled to nitrite reduction is produced by members of the phylum *Rokubacteria*. These bacteria have also been found in Pindal Cave [42] and in an Alpine cave [43].

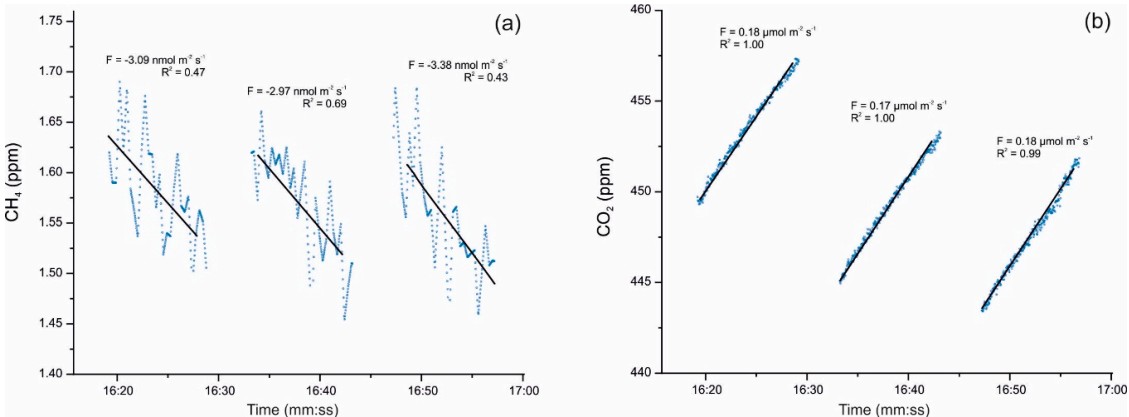

**Figure 2.** (**a**) Detail of $CH_4$ uptake fluxes with an average of $-3.15$ nmol m$^{-2}$ s$^{-1}$ and (**b**) simultaneous $CO_2$ emission fluxes uptake fluxes with an average of $+0.17$ μmol m$^{-2}$ s$^{-1}$, monitored on 17 December 2019 directly above sediments inside Pindal Cave (Asturias, Spain). The value of the diffusive flux (F) and the corresponding exponential adjustment ($R^2$) of each measurement are indicated.

Other studies combining the depletion of $CH_4$ with other GHGs ($N_2O$ and $NO_2$) were carried out in Vapor Cave, Southeast Spain. This is a hypogenic cave formed by the upwelling of hydrothermal $CO_2$-rich fluids in which anomalous concentrations of nitrogen oxides can be found [44]. The cave is characterized by a combination of rising warm air with large $CO_2$ outgassing and highly diluted $CH_4$ of endogenous origin. Additionally, extreme environmental conditions were observed, such as high air temperatures (38–43 °C) and 100% relative humidity, hypoxic conditions (17% $O_2$), $CO_2$ concentrations that exceed 1%, $^{222}$Rn activity with values above 50 kBq/m$^3$, and a vertical thermal gradient of 3.2 °C/100 m [45]. These conditions, associated with the combined effects of tectonic activity and hydrothermalism, make this cave a remarkable site for the study of uncommon or extremophilic microbial communities. In Vapor Cave, the depletion of $CH_4$ was quantified to account for more than 60% removal of the deep endogenous component of this gas [45].

Martin-Pozas et al. [44] collected different cave air and sediment samples from −2 to −80 m in Vapor Cave. The analyses were conducted by taking advantage of technological advances in high-precision field-deployable CRDS and FTIR spectrometers, which allowed to measure target tracer gases ($NO_2$, $N_2O$, $CH_4$, and $CO_2$) and $\delta^{13}C$ of both carbon-GHGs in situ. The $\delta^{13}C_{CO2}$ data (−4.5 to −7.5‰) suggested a mantle-rooted $CO_2$ likely generated by the thermal decarbonation of underlying marine carbonates, combined with degassing from $CO_2$-rich groundwater. $CH_4$ molar fractions and their $\delta D$ (−77 to −48‰) and $\delta^{13}C$ values (−52 to −30‰) indicated that the $CH_4$ reaching Vapor Cave is the remnant of a larger and deep-sourced $CH_4$, which was likely generated by the microbial reduction in carbonates. This $CH_4$ was affected by a postgenetic depletion during its migration through the cave environment as a component of the rising warm air.

$CH_4$ concentrations and $\delta^{13}CH_4$ varied with depth. At −80 m, higher concentrations were found but above −30 m depth lower $CH_4$ concentrations were found and heavier $\delta^{13}C$ values were found near the cave entrance. This was consistent with a methane oxidation mediated by microorganisms and in fact, next generation sequencing (NGS) analysis of sediments showed a relative abundance of *Candidatus* Methylomirabilis 4 to 5 times higher in the deepest sample (−80 m) with respect to −30 and −15 m. *Candidatus* Methylomirabilis oxyfera (*Rokubacteria*) is an anaerobic denitrifying methanotroph [46]. It must be noticed that Isobe et al. [47] found that members of the uncultivated candidate phylum *Rokubacteria* responded positively to elevate $CO_2$ concentrations.

In a similar way, Cappelleti et al. [48] studied an area of agricultural soils in Italy with anomalously high temperatures (up to $\cong$ 50 °C) and found emissions of biogenic $CO_2$ linked to $CH_4$ oxidation at a depth of 0.7 m from the surface. A strong biological methane-oxidizing activity in these soils was found and an examination of the *pmoA* clone libraries revealed the large biodiversity of methanotrophs including *Methylomonas, Methylococcus, Methylocystis*, and *Methylocaldum*.

Regarding the nitrogen gases, Martin-Pozas et al. [44] stated that the analysis of the ecological functions and metabolism of the microbiota from cave sediments suggested that $N_2O$ is mainly produced in the deepest areas of Vapor Cave (below −15 m depth). In these areas, high $CO_2$ concentrations and low $O_2$ levels within the sediments determine a prevailing hypoxic and acidic environment that promotes the release of nitrite, nitric oxide, and hydroxylamine as products of the metabolism of ammonia-oxidizing archaea and nitrate reduction. In fact, at −15 m depth, the archaeal communities were dominated by the class *Nitrososphaeria* (69.0% of the total *Archaea*), with a majority of uncultured members and only two identified genera, *Nitrososphaera* and *Nitrosotenuis*. This is consistent with the abundant occurrence of these *Archaea* in deep sediments and better survival under conditions of low dissolved oxygen.

To summarize, considerable advances have been reached in recent years regarding processes of production, consumption, and storage of greenhouse gases ($CO_2$, $CH_4$ and $N_xO_x$) by cave microorganisms in subterranean vadose ecosystems. Recent and current research has shown that cave *Actinobacteria* are active agents in the fixation of $CO_2$, capturing $CO_2$ from air and forming calcium carbonate polymorphs [8]. In particular, direct $CO_2$ flux measurements in areas heavily colonized by bacteria indicate that they were promoting the uptake of this gas. Subterranean environments act as sinks or net sources of soil-derived carbon dioxide ($CO_2$) on annual and even daily scales, reaching up to ten times higher than the mean atmospheric $CO_2$ content, which involves the exchange of large amounts of $CO_2(g)$ with the lower troposphere and its role as a depot and/or emitter. In a very recent in-situ experimental work (Pindal Cave, Spain) with a closed chamber-based gas exchange system—research in progress—we have verified negative $CH_4$ fluxes (uptake) from microbial communities, simultaneously linked to positive $CO_2$ fluxes (emission) directly related to microbial methanotrophy. The most recent data from direct measurements of gas exchange fluxes indicate that both gases are inextricably linked in these microbial-induced processes.

### 3. The Search of Antibiotics Produced by Subsurface Bacteria

Methanotrophic bacteria not only perform consumption of $CH_4$ in caves, but also have other capabilities. Methanotrophs produce methanobactins, bioactive compounds which have a high affinity for copper, but can bind additional metal ions, suggesting that these compounds might play a role in the protection against toxicity of metal ions other than copper [49]. Methanobactins exhibited antibiotic activity against Gram-positive bacteria and have been investigated as a treatment for Wilson disease, a human disorder involving toxic copper accumulation. For more detailed information, the reader is referred to the review of Kenney and Rosenzweig [49] (and references therein). Other uses of methanotrophs rely on their application and potential value in bioremediation, namely methane removal from landfills and coal mines, as well as biodegradation of toxic compounds [50,51].

In the biosphere, methanotrophs share niches with other bacterial groups, namely *Proteobacteria* and *Actinobacteria* [43,44]. In fact, several reports have shown the interactions between methanotrophs and heterotrophs and clear beneficial associations promoting growth were observed among them [52–54]. Most of these heterotrophs are well-known producers of bioactive compounds (antibiotics).

### 3.1. Why Is There a Need of New Antibiotics?

According to the World Health Organization (WHO) [55], there is a serious lack of new antibiotics to combat the growing threat of antimicrobial resistance of pathogenic bacteria as well as an urgent need for more investment in research and development to fight against antibiotic-resistant infections. De Kraker et al. [56] predicted more than ten million deaths of people infected with the antibiotic-resistant bacteria worldwide within the next 30 years if urgent measures are not taken now.

In 2017, the WHO [57] reported a list of 12 bacteria for which novel antibiotics were urgently required. The list comprises three categories: critical, high and medium priority, according to the urgency of need for new antibiotics. In critical priority were included *Acinetobacter baumannii, Pseudomonas aeruginosa,* and *Enterobacteriaceae.* High priority was assigned to *Enterococcus faecium, Staphylococcus aureus, Helicobacter pylori, Campylobacter* spp., *Salmonellae,* and *Neisseria gonorrhoeae,* and finally the category of medium priority encompasses *Streptococcus pneumoniae, Haemophilus influenzae,* and *Shigella* spp. It is remarkable that only three out of the twelve bacteria from the list are Gram-positive, which denotes the interest in having antibiotics active against Gram-negative bacteria.

The clinical pipeline for new antibiotics currently includes only a small number of novel compounds in development. In the past 20 years, only two new antibiotic classes, both only active against Gram-positive bacteria, have received global regulatory approval by the US Food and Drug Administration and European Medicines Agency [58]. In the same time period, no new antibiotics against Gram-negative bacteria have been approved, and the quinolones, discovered in 1962, represent the last novel drug class identified to be active against Gram-negative bacteria. In addition, only two completely new drugs for multidrug-resistant tuberculosis treatment have reached the market in over 70 years [58].

Currently, the clinical pipeline is dominated by improvements of existing products with similar chemical structures, but some level of cross-resistance and fast adaptation of target bacteria can be expected. Resistance to one specific antibiotic agent can lead to resistance to a whole related class, through exchange of genetic material between different bacteria, and can affect antibiotic treatment of a wide range of infections and diseases [59]. Ideally, research should be focused on entirely new classes of compounds, targets, and modes of action to avoid cross-resistance to existing antibiotics [60].

### 3.2. Antibiotics from Subsurface Bacteria

About two-thirds of all known antibiotics were produced by *Actinobacteria*, particularly by species of the genus *Streptomyces* [61,62]. Historically, the decades from the 1930s to 1980s represented the golden age of antibiotic discovery. The starting point was the exhaustive screening on soil *Actinobacteria* carried out by Waksman and collaborators, which led to the discovery of actinomycin and streptomycin [63,64].

Further screenings of soil bacteria produced thousands of bioactive compounds, such as the well-known chloramphenicol, tetracycline, erythromycin, vancomycin, gentamicin, etc. However, in subsequent searches the same or similar compounds already known were recurrently obtained from soil bacteria. Even different species produced the same compounds [65]. Therefore, the investigations adopted different strategies.

A few authors stated that new bioactive compounds can be found in *Actinobacteria* that have been previously studied to the extent that re-examination of known microorganisms, already in storage, should provide novel compounds [66,67]. In this context, Takahashi and Nakashima [67] tested lyophilized actinobacterial strains from the Kitasato University Microbial Library, Japan, most of them more than 35 years old, and found that 330 strains were producers of useful bioactive compounds. In this work, a strain of *Streptomyces griseus*, isolated from a soil sample in 1971, and producing streptomycin, was cultivated in four different media and revealed to yield two new N-containing compounds, iminimycin A and B, both with antimicrobial activity against Gram-positive and Gram-negative bacteria. This finding was possible due to the methodology adopted, a physico-chemical screening of culture broths involving the use of a reagent to identify nitrogen-containing metabolites and a routine analysis by liquid chromatography–mass spectrometry, liquid chromatography–UV detection, and polarity and their comparison with existing databases.

It is amazing that after Schatz et al. [64] discovered streptomycin in a *S. griseus* strain, about 200 compounds have been reported from other strains identified as *S. griseus*, and still new secondary metabolites are being discovered, pointing to *Actinobacteria* as an inexhaustible source of naturally occurring antibiotics [67].

In a search for bioactive compounds, some authors moved to different or unexplored ecosystems other than soils. In this respect, the exploration of marine organisms is conducted to discover novel bioactive compounds—such as two nucleosides extracted from the sponge *Tectitethya crypta*: spongothymidine and spongouridine [68], or more recently trabectedin, obtained from the tunicate *Ecteinascidia turbinata* [69]. Another approach was to increase the efforts in screening rare *Actinobacteria* genera [70] or explore unknown niches [71].

In the biosphere there is a little-explored niche which can provide promising results: the subterranean environment. The study of the microbiology of caves is an interesting line of research and allows the search of a great diversity of unknown bacteria and fungi and the possible production of new bioactive compounds, as reported by Cheeptham [72] and Cheeptham and Saiz-Jimenez [73].

Caves are colonized by complex bacteria communities and are an excellent reservoir for new species of *Actinobacteria* [71,74,75]. Research on this topic has not been considered with all the dedication it deserves, to the point that the microbiology of most caves is relatively unknown.

Nowadays, two main approaches are adopted in the study of cave bacteria. One is the isolation of the organism and the subsequent assay of the antimicrobial activity of extracts against pathogenic bacteria. Numerous studies can be found in the literature, for which only a few representative papers are cited here [76–81]. In most papers, attempts to identify the bioactive compounds through chemical and structural analyses were not accomplished. The screening of only antimicrobial activity without a structural elucidation of the involved metabolites may not be useful for the discovery of new antibiotics.

A few examples of studies where the bioactive compounds were fully identified are those of Herold et al. [82], who provided the chemical structure of cervimycin A-D, a polyketide glycoside complex obtained from *Streptomyces tendae*, isolated from Grotta dei Cervi, in Italy, and the huanglongmycin A-C complex, synthesized by a strain of *Streptomyces* found in a Chinese cave [83].

Axenov-Gibanov et al. [84] isolated a *Streptomyces* sp. from a cave moonmilk which produced cyclodysidin D and chaxalactin B. Another 120 metabolites were observed in the liquid extract, from which a total of 102 compounds could not be identified and appeared to be novel. These three cases exemplify how caves are a research field with high potential for novel drug discovery.

In addition to conventional isolation techniques, several authors explored the possibilities of increasing the recovery of novel or interesting bacteria by improving isolation and cultivation

techniques in order to extend the number of bacteria producing bioactive compounds under laboratory conditions [85]. Some of these methodologies include pretreating samples under different conditions—air drying, dry heating, moist incubation, desiccation, setting different pH, design of new culture media mimicking nature conditions, etc.—for an effective isolation of rare *Actinobacteria* [71,86,87]. A list of non-specific and specific methods that enhance the production of bioactive compounds can be found in Manteca and Yagüe [88].

Another approach is represented by NGS techniques in combination with genome mining which revolutionized the field of antibiotic research. According to Nett et al. [89] genome mining analyses suggest that less than 10% of the genetic potential of antibiotic producers is currently being used, which indicates that there is a huge untapped genetic reservoir waiting to be exploited for drug discovery.

Currently, more than 1555 completed genome sequences of *Streptomyces* are available in EzBiocloud [90] and, for instance, *Streptomyces coelicolor* harbors 22 secondary metabolite gene clusters but really produces only four of the encoded metabolites under standard laboratory conditions [58].

Screening for biosynthetic genes is an effective strategy to characterize bioactivity. For Bukelskis et al. [91] information on the expression of biosynthetic genes encoding for various bioactive compounds in cave bacteria is either limited or missing, and genome mining for PKS and NRPS genes in parallel with transcriptional analysis of the identified genes would be the more effective strategy to analyze and exploit the bioactivity of cave bacterial strains.

Along with genomic mining, the combination of culturing techniques and transcriptomics would complement the systematic investigation for bioactive compounds in bacteria. These techniques have been recently used to investigate the production of the antibiotic andrimid in the marine bacterium *Vibrio coralliilyticus* [92]. In this study, the authors reported the differential expression of five biosynthetic gene clusters as well as an overproduction of andrimid in the presence of chitin rather than glucose in the culture medium. Alteration of cultivation parameters, such as, solid/liquid culturing, the presence/absence of nutrients, variations of pH and temperature, or changes in aeration supplying, would lead to the activation/inactivation of metabolic pathways involved in the biosynthesis of antimicrobials. These conditions or phenotypic variations are used in transcriptomic analyses to identify the clusters of genes differentially expressed in every condition, and afterwards, expressed in heterologous hosts by means of vectors such as plasmids, cosmids, fosmids, and artificial bacterial chromosomes or bioactive compounds. Methodology based on heterologous expression and subsequent screening has been widely implemented in several species of the genus *Streptomyces* and other actinobacteria for secondary metabolism expression and subsequent identification of bioactive compounds [93].

Culturing and isolation of microorganisms has been the primary methodology in new antibiotics discovery but, although nowadays it is extensively used, this approach is biased by the impossibility of extrapolating the subsurface microbiome to the laboratory. In fact, the majority of bacterial species cannot grow in laboratory conditions, an issue that could have led to disregarding a huge amount of new antibiotics synthetized by uncultured bacteria [87]. To amend the culture-dependent bias, metagenomic mining and metatranscriptomics should raise the chance to acquire novel bioactive compounds directly from the subterranean ecosystems.

Metagenomics allows the study of genomes from non-culturable bacteria by means of sequencing the in-situ collected DNA and the subsequent bioinformatic analyses for assembly, binning, and annotation of the genomes of bacteria present in the sample. As a result of the treatment of metagenomic data, potential biosynthetic gene clusters are set in libraries and expressed by heterologous hosts to evaluate and validate their bioactivity.

Metatranscriptomics have a double impact in microbiome studies—on one hand, RNA sequencing can focus on the metabolically active bacteria, discriminating the ancient DNA and latent or inactive bacteria that could add "noise" to the study, since the variation of relative abundance and presence of bacterial communities has been observed, even at the phylum level, in rRNA 16S studies when

comparing total and metabolically active communities using both cloning and NGS analyses [33,94]. Thus, efforts of bioactivity analyses could be more exhaustive and accurate. On the other hand, interactions inter- and intraspecies are the best frames to investigate the bioactivity of bacteria in competition with other species sharing the same niche.

In the last decade, a few projects have been funded by the European Commission and National Organizations. They are SeaBioTech (https://spider.science.strath.ac.uk/seabiotech/index.php), Marex (https://www.marex.fi/), PharmaSea (http://www.pharma-sea.eu/pharmasea/), FucoSan (https://www.fucosan.eu/en/), Tascmar (https://cordis.europa.eu/project/id/634674), and Probio (https://www.vliz.be/en/news?p=show&id=8386). All of them focused their activities on marine organisms (bacteria, algae, invertebrates, etc.). Activities and results can be found in their respective web pages. As far as we know no project on terrestrial microorganisms is ongoing, other than a recent research project launched with the aim of studying the biodiversity of extreme environments such as abandoned and active mines in Euroregion A3 (Alentejo, Algarve, and Andalucia) [95].

The mines (pyrite, manganese, copper, etc.) to be investigated are located in the Iberian Pyrite Belt. The Iberian Pyrite Belt is one of the World's largest accumulations of mine wastes and acidic mine waters from drainages, which have caused severe pollution by the low pH and presence of dissolved metals. This acidity and metal pollution has caused the loss of most forms of aquatic life, with the exception of acidophilic microorganisms which inhabit these extreme environments [96]. In this scenario, microorganisms are subjected to stress and the need to develop a metabolic system capable of coping with oligotrophy (lack of organic nutrients) and the expression of genes that produce bioactive compounds to compete with other organisms for the scarce nutrients available in the acidic environment.

The inclusion of mines in this project represents a further step and an innovative research area, since as far as we know there are no previous reports considering mine microbiomes as a source of bioactive compounds.

Preliminary results (Table 1) showed that about 14% of the bacteria isolated from caves and mines produced bioactive compounds against the tested pathogens. It was noteworthy that bacteria isolated from sediments of two submarine cave and marine organisms inhabiting the caves, located in the Algarve coast, only yielded two positive strains, in contrast with the higher number of terrestrial isolates. In addition, it is remarkable that 29% of bacteria isolated from the Iberian Pyrite Belt mines showed inhibitory activity against pathogens, which doubled the percentage of cave bacteria. Most of the bacteria showed high activity against Gram-positive bacteria and lower against Gram-negative bacteria. This is a reason to focus the search on having antibiotics active against Gram-negative bacteria, as demanded by the WHO.

**Table 1.** Screening of bacteria from mines and marine caves in Alentejo, Algarve, and Andalucia, as well as from the IRNAS-CSIC cave bacterial collection.

| Origin | Tested Strains | Positive Strains | 1 | 2 | 3 | 4 | 5 | 6 |
|---|---|---|---|---|---|---|---|---|
| Terrestrial caves | 863 | 126 | 77 | 48 | 116 | 18 | 31 | 24 |
| Marine caves | 144 | 2 | 0 | 0 | 2 | 0 | 1 | 0 |
| Mines | 79 | 23 | 13 | 7 | 14 | 3 | 8 | 3 |
| Total (bacteria) | 1086 | 151 | 90 | 55 | 132 | 21 | 40 | 27 |
| Total (%) | 100% | 13.9% | 8.29% | 5.06% | 12.15% | 1.93% | 3.68% | 2.49% |

1: *Bacillus cereus* CECT 148; 2: *Staphylococcus aureus* CECT 4630; 3: *Arthrobacter* sp. LR584284; 4: *Pseudomonas aeruginosa* CECT 110; 5: *Escherichia coli* DSM 105182; 6: *Acinetobacter baumannii* DSM 300007.

A complete screening of bacteria from air, water, and sediments from mines provided a bacterium collected with an air sampler [97] inside the mine of Lousal (Portugal). Based on the study of the 16S rRNA gene and subsequent sequencing of the genome, the strain was identified as a species within the genus *Pseudomonas* and showed a high bioactivity against all the tested pathogenic

bacteria. Genome mining analysis focused on the secondary metabolism with antiSMASH [98] which resulted in the prediction of 19 biosynthetic gene clusters from different domains such as polyketide synthases, non-ribosomal peptide synthases, tRNA-dependent cyclodipeptide synthases, aryl polyenes, phenazines, bacteriocins, N-acetylglutaminylglutamine amides, betalactones, flavin-dependent tryptophan halogenases, homoserine lactones, and siderophores (Figure 3).

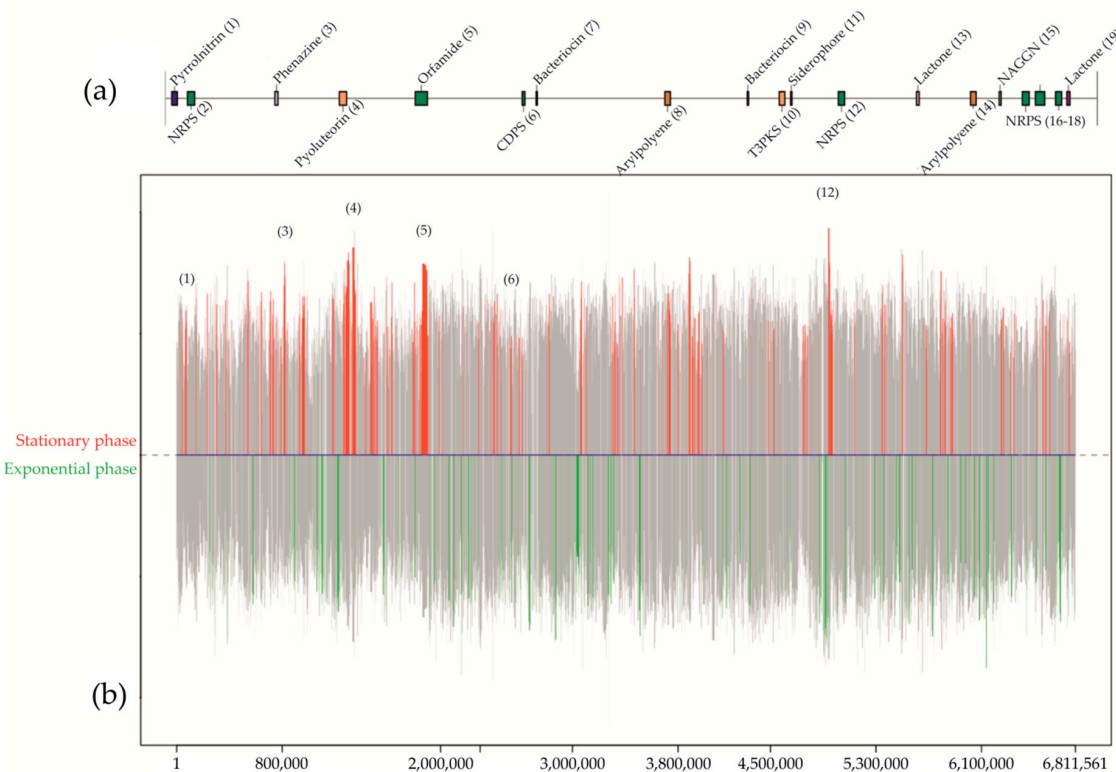

**Figure 3.** Graphic example from a differential gene expression analysis for *Pseudomonas* sp., isolated from a mine in Lousal (Portugal). Culturing conditions were established in solid medium with nutrient agar and 2% of glycerol. The prediction and location of the biosynthetic gene clusters in the 6.8 Mbp genome (**a**) were used to map the differentially expressed genes (**b**) between the stationary phase and the exponential along the chromosome. Colored peaks represent the variability of expressed genes between both conditions with a 99% of probability. Red peaks show those genes differentially expressed in the stationary phase, in opposition to the green peaks, which are referred to the differential expression in the exponential phase. Grey peaks represent those genes differentially expressed below the 99% threshold, or simply not differentially expressed between conditions. From the predicted secondary metabolites, only six of them were differentially expressed in the stationary phase. Considering the bioactivity analyses in this medium and the study of antibiotic resistance mechanisms on genomes of pathogenic bacteria, no expression of genes involved in the synthesis of antibiotics against Gram-negative bacteria occurred for the described solid media culture.

An "One Strain Many Compounds" (OSMAC) approach [99] was used to check the bioactivity of *Pseudomonas* sp. with the aim of performing a comparative analysis of gene expression by means of transcriptomics techniques. A variation of pattern in the inhibition of pathogens was observed between solid and liquid cultures. The bioactivity of *Pseudomonas* sp. was limited to Gram-positive bacteria in a solid culture with nutrient agar and 2% of glycerol, whereas for the liquid version of the same culture, the inhibition of pathogens was total, for both Gram-positive and Gram-negative bacteria. Inhibition of growth in pathogens was only observed during the stationary phase. Thus, prediction of gene clusters involved in the synthesis of secondary metabolites, comparison of analyses based on the gene expressions, and the study of the antibiotic's resistance mechanisms, or resistome, from the

pathogenic bacteria genomes allowed the identification of specific genetic mechanisms involved in the antimicrobial bioactivity for the tested *Pseudomonas* sp. strain.

For a correct management and understanding of genomics and transcriptomics data, advances of bioinformatics have empowered the development of these omics sciences. Software tools addressing quality checking, correction, assembly, mapping, and annotation of data represent the basic procedure, but the availability of hardware with the capacity to carry out high-level operations is essential. However, these analyses and processes are harder and tedious for functional metagenomics and metatranscriptomics because of a higher presence of data coming from hundred or even thousand bacterial genomes present in the same sample. In this sense, the continuous improvement of bioinformatics tools and workflows are needed to achieve a deeper understanding of biological processes [100].

The improvement in NGS platforms and the rising use of omics sciences has stimulated the falling cost of sequencing and the appearance of better computational methodologies. Nevertheless, although these techniques are presented as a prominent stimulus in new antibiotics research, few studies have focused on cave microbiome have used metagenomic and metatranscriptomic analyses so far. Beyond the inherent bias to the application of nucleic acid-based techniques such as a low concentration of DNA/RNA, partial fragmentation, and presence of enzyme inhibitors [101], current limitations are focused on the training of computational scientists to analyze the complex data as well as sequencing enough samples to get a powerful study [102].

## 4. Conclusions

Subterranean ecosystems constitute a huge subsurface reactor of the global biogeochemical cycle with a potential and regular buffering effect on long-term increments of atmospheric GHGs linked to climate change. The subterranean microbiome plays a primary regulatory role in its gas composition, controlling the uptake-fixation-production of $CO_2$, $CH_4$ and $N_xO_x$ gases, as well as their coupled evolution during their migration through the critical zone to the lower troposphere. Recent studies apply an innovative and multidisciplinary combination of a broad suite of cutting-edge technologies -GHG flux monitoring, isotopic geochemical tracing, biogeochemistry, metagenomics, etc.—to quantify the GHG fluxes controlled by microbial-induced processes and directly exchanged with the cave atmosphere in several temporal scales (daily, seasonal, annual pattern). The next step should be to establish the feedback mechanisms between environmental-microclimatic conditions and rates and type of activity of microbial communities in subterranean ecosystems.

The recent in-depth study on the microbial activity of the subterranean microbiota has not only provided qualitative and quantitative data on the regulation of the concentration of $CO_2$ and $CH_4$, but has also shown that methanotrophs and heterotrophs can interact and stimulate the growth of each other. This growth results in the production of bioactive compounds. Although the literature describes the isolation of many cave bacteria with inhibitory properties against pathogens, only a few studies provide the identification and chemical structure of the metabolites. The advent of massive sequencing technologies or NGS has promoted the development of the so-called omics sciences, offering a holistic view of the inter- and intracommunity relationships of microorganisms. Genomics, transcriptomics, and the consequent massive generation of data have favored the development of bioinformatics as an essential interdisciplinary field, in continuous progress, for the interpretation of biological processes. This set of techniques and disciplines has not only allowed a more exhaustive knowledge of the biology of microorganisms, but it has made possible to overcome the barrier of extrapolating the ideal conditions for microbial growth in the natural environment to the laboratory, and in this way it has also been made possible to identify new genetic mechanisms involved in the synthesis of bioactive compounds.

**Author Contributions:** Conceptualization, S.S.-M., S.C. and C.S.-J.; investigation, T.M.-P., S.C., V.J., J.L.G.-P., I.D.-M. and J.C.C.; writing—original draft preparation, T.M.-P., S.S.-M., J.L.G.-P., A.F.-C. and C.S.-J.; writing—review and editing, C.S.-J. All authors have read and agreed to the published version of the manuscript.

**Funding:** Financial support was obtained through project 0483_PROBIOMA_5_E, co-financed by the European Regional Development Fund within the framework of the Interreg V-A Spain-Portugal program (POCTEP) 2014–2020. This work was also supported by the Spanish Ministry of Economy and Competitiveness through projects CGL2016-75590-P and PID2019-110603RB-I00, AEI/FEDER, UE.

**Conflicts of Interest:** The authors declare no conflict of interest.

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
