# Peer review of "Microbial Activity in Subterranean Ecosystems: Recent Advances"

_applsci, doi:10.3390/app10228130_

Round 1

Reviewer 1 Report

Comments to the Authors:

Manuscript Number: ID: applsci-988018

Title: Microbial activity in subterranean ecosystems: Recent advances

Overview:

The bacterial activity in subterranean ecosystems is an important aspect for the environmental microbiology field. Notwithstanding the several literature gaps, it is evident that the microbial community play an important role in many ecosystem processes.The manuscript is well written but some minor revisions are necessary to improve it.

Minor revisions:

line 64: replace “are the main process” with “are the main processes”

Line 68: the authors provide references for the first part of the sentence related to bio-induced mineral formations in subterranean environments. However, no references have been reported for the second part related to the microbial-rock interaction related to the CO2 uptake or release processes. Please, specify if these references are suitable for both parts or provide them.

From line 79 to 84: this is a very long sentence that makes difficult read the manuscript. Furthermore, in the line 82, the authors reported a reference for each research field with the exception of biotechnology sector. The authors should split out this sentence providing a reference for the biotechnology field.

line 85 and 86: replace 1. “The control…” with (1) “the control…” as well 2. “The search..” with (2) “the search..”

Line 130: replace it with its

Line 132: the authors should provide the extension of pmoA and a brief explanation of particulate methane monooxygenase genes.

Line 136: replace with “These soil bacteria oxidize the atmospheric CH4

From line 144 to 147: no reference was reported

From line 189 to 195: the authors should better explain the Figure 1 in the text, especially the data about the figure 1b and 1c.

Line 287: the title of this paragraph indicates a section describing the research progress about the bioactive compounds produced by subsurface bacteria. However, the first part, especially from the line 288 to 327, is a summary of general information about the antibiotic-resistance and it is not of interest for the reader in this context. Moreover, this section is too long and it could be divided in two different parts: the first one related to the novel bioactive compounds and the second describing all of the approaches adopted to study the cave bacteria.

Table 1: improve the organization of this table. For example, the names of bacterial species result truncated as well the bracket of total bacteria is not aligned with the rest.

Line 428: replace with Metagenomics allows

Line 466: introduce a blank space between the table 1 and the starting point of this paragraph to facilitate the reader

A section with the conclusions should be added because this review is a wide collection of information and a final view of the best evidences about this topic should be provided.

Reviewer 2 Report

General comments:

The manuscript is a well-written review of some recent advances of the research on microbial activity in subterranean ecosystems - caves.
The focus is on the influence of the microbial metabolism on greenhouse gases, especially on methane and carbon dioxide,
and the potential of cave microbiomes for the discovery of novel antibiotics.
These topics are, as pointed out in the manuscript, emerging research topics
that have potential and this review might well provide a basis to define such research topics.
However, the two focal topics only become clear from the abstract and from the title only,
I expected a more comprehensive discussion on microbial activity in subterranean ecosystems.
Having stated that, I still think such an all-encompassing, comprehensive discussion on microbial activity in subterranean ecosystems
is quite challenging and focusing on some important subtopics is a good idea.

A general comment I have is that I find the two parts of the manuscript have little to no connection,
besides the location of greenhouse gas metabolism or origin location of the potential novel antibiotics.
I would almost like to suggest to split this review into two separate manuscripts, each focusing on one topic exclusively.
At least, the authors should provide some reasoning why they limit their focus on these two interesting,
but seemingly randomly chosen topics and discuss them in a single manuscript.
Also, the antibiotics part is much underrepresented in the abstract, which, except for the last quarter,
focuses exclusively on greenhouse gases and the impact of subterranean microbes.

Some more minor general comments are that in the abstract and the introduction, the authors are, in my opinion,
a bit too unspecific in their use of karst/karstic and cave, using the terms somewhat interchangeably.
I think that while caves are very common in karst and a prominent feature of karstic environments,
there are other forms and origins of caves as well.
One example of non-karstic caves are primary caves formed at the same time as the host rock in e.g. cooling lava.
Further, the authors could, in my opinion, elaborate more on why caves and mines are
one of the previously overlooked but promising places to search for microbes producing yet undiscovered, novel antibiotics.
There is some discussion in the lines 354-355, 361-362, and 366-368,
but I personally am not fully convinced by the argumentation there.
They also could elaborate more on the potential of subterranean microbes for the discovery of novel antibiotics.
Of the 5 pages on the topic in the section, only 36 lines (361-383, 428-440) are actually focused on the "core" topic,
together with an oddly specific overview for 2.5 pages of a single project, that the authors are involved in (see the funding acknowledgement).
If this truly is the unique and only research related to the topic, this should be mentioned and somehow introduced.
The parts do not feel that well connected.
The rest, 2-2.5 pages, of section 3 is general introduction, problem and methodology description.

Specific comments (line, table or figure number):

41: "Shallow karst ecosystems cover up to 25% of the Earth’s land surface" Do you have a reference for that?

44-45, 48-51: " as well as absence of light and scarcity of nutrients. However, this is not a limitation for microorganisms", "microorganisms are forced to adapt their metabolism for surviving in extreme conditions, and the low input of carbon, nitrogen and phosphorus as well as the chemical composition of the rock has a direct impact on the community diversity."
Those two statements are completely contradictory, or at least not well formulated. I would suggest reformulating the statement in line 45 in a way that makes it clear that this does not exclude or inhibit microbial life, in general. This should remove the contradiction.

72-74: "Particularly, it is poorly understood the functioning of microbe interactions with air-water-rock interfaces in subterranean ecosystems, and the chemical and biological processes that the microorganisms carry out and their response to environmental triggers." This reads somewhat strange, consider rewording.

75-87: Here, I miss a brief description, why the authors chose from their list of interesting research topics related to subterranean
microorganisms, the two topics they wrote this review about.

75-76: Why are microorganisms so especially essential in subterranean ecosystems?

107-112: I do not get the reasoning behind this. Where is the logical connection between " A CH4 sink represented by caves and other shallow vadose environments has been largely overlooked in the scientific literature." and "This means that methanotrophs play a role in the ecosystem"? Do you mean something like: "Caves and other shallow vadose environments are home to methanotrophic microorganisms and thus represent a CH4 sink. This subterranean CH4 sink is largely overlooked in the scientific literature"

144: "However, further research verified that the mechanism of CH4 consumption" research by whom? Still Fernandez-Cortes et al. [26]?

273 and following: Indeed, to measure the impact of the subterranean microbial CH4 sink on the atmosphere and global climate (change), one would need to measure the fluxes of the greenhouse gases exchanged between atmosphere and the subterranean environments. Showing that there is a consumption of e.g. CH4 in caves is just an the initial step.

293-295: " Strathdee et al. [44] stated that antimicrobial resistance might worsen under COVID-19 due to their association with bacterial infections and the overuse of antibiotics in patients even when not clinically indicated." Is this statement really relevant in this context? Are there no more general examples of miss use of antibiotics leading to the build-up of resistant pathogens? I am sure there are. I feel COVID-19 is simply mentioned here for no immediate scientific reason and more to sneak a currently popularly searched keyword in.

296-306: Is this detailed list of pathogenic bacteria for which novel antibiotics are needed really necessary for this manuscript? Would the first and last sentence of the paragraph be enough?

447: What stresses exactly are the microorganisms exposed to? What are the differences of mine environments to cave environments?

tab. 1: It might be interesting to split the "Mines and caves" origin into separate "Mines" and "Caves". Especially if there is a significant difference between the microbial communities in mines and caves, see also my comment to line 447.

fig. 3: I do not understand. Aren't the color-coded occurrences in 3b expressions of genes involved in the synthesis of antibiotics? If not, what are they that the grey background is not? Aren't some genes even identified with 3a, or what are the small numbers referring to? I am very confused by either the figure or the figure's caption.

Round 2

Reviewer 2 Report

The revision improved the manuscript significantly and I feel that publication in Applied Sciences should now be possible.

I would have preferred if the authors had adapted the manuscript title to be more specific as I hinted at in the 1st round of review already, but I don't want to be too pedantic about that. For me the current title is a bit misleading and can be understood to promise more than actually is detailed in the manuscript, but it is technically anyway still correct.
My last comment to the authors would be that while the updated lines 462-482 relate to many of my initial comments and questions, they didn't match in my opinion in all the occurrences in their response to my comments or questions. However, the revision of the manuscript do not focus exclusively on those 20 lines and thus also addressed those comments.

To summarize again, I think that the revised version should be ready to be published.